# Clinical Features of Ocular Motility in Idiopathic Orbital Myositis

**DOI:** 10.3390/jcm9041165

**Published:** 2020-04-18

**Authors:** Min Seok Kang, Hee Kyung Yang, Namju Kim, Jeong-Min Hwang

**Affiliations:** 1Department of Ophthalmology, Kim’s Eye Hospital, Seoul 07301, Korea; nietzsche@khu.ac.kr; 2Department of Ophthalmology, Seoul National University College of Medicine, Seoul National University Bundang Hospital, Seongnam, Gyeonggi-do 13620, Korea; nan282@snu.ac.kr

**Keywords:** strabismus, idiopathic orbital inflammatory syndrome, orbital myositis, extraocular muscle, imaging findings

## Abstract

Objective: To elucidate the clinical features of ocular motility and the risk factors for recurrence in idiopathic orbital myositis. Methods: The medical records of 31 patients diagnosed with idiopathic orbital inflammation between 2003 and 2019 were retrospectively reviewed. All patients were initially treated with corticosteroids. Treatment outcome and ocular motility were noted. Results: Twenty-six patients (84%) had unilateral involvement and five patients (16%) were bilateral. Of the 31 patients, 22 patients (71%) showed ocular motility limitation. The mean grading scale of extraocular muscle (EOM) limitation was −1.65 ± 1.80. EOM limitation was found in the same direction of the most affected muscle in 14 patients (64%), while 8 patients (36%) showed duction limitation in the opposite direction. Nine patients (35%) suffered from recurrence. Recurrence was more likely to occur in patients with multiple muscle involvement (*p* < 0.001). The interval to relapse of symptoms after discontinuation of steroids was significantly shorter in patients with multiple recurrences compared to those with a single recurrence (1.8 ± 0.8 weeks versus 6.0 ± 1.4 weeks, *p* = 0.020). Conclusions: Idiopathic orbital myositis showed variable degrees of ocular motility limitation, and limitation in the same direction of the action of the affected muscle was more frequent. Recurrent myositis was more likely to have multiple muscle involvement. Rapid relapse of symptoms after discontinuation of steroids was a significant indicator of multiple recurrences.

## 1. Introduction

Idiopathic orbital inflammatory syndrome (IOIS), previously referred to as pseudotumor, is a broad term covering all inflammatory diseases that influence the structures contained within the orbit external to the ocular globe [1,2,3]. IOIS has highly variable clinical features targeting diverse orbital tissues, such as the lacrimal gland, extraocular muscles (EOM), and orbital tissues [1,2,3,4,5,6]. The diagnosis of idiopathic orbital inflammatory syndrome is challenging because of the clinical diversity, i.e., from being acute to chronic, mild to severe, isolated to associated with systemic diseases, number of the affected muscles, presence of symptoms, or recurrences [1]. Diagnosis is based on patient’s history, clinical manifestations, and typical features on computed tomography (CT) or magnetic resonance (MR) imaging. The main therapy consists of systemic corticosteroids, but other options including external beam radiotherapy, antimetabolites, immunosuppressant agents, and surgical debulking have also been used [4].

Orbital myositis is a rare and relatively rapid onset disease, which represents a subgroup within the diagnostic entity of IOIS, resulting in inflammation mainly of the EOM. There have been many case reports of orbital myositis or orbital inflammation [7,8,9,10,11,12,13,14,15,16,17,18,19,20]. However, a large scale case series depicting the clinical manifestations of ocular motility limitations associated with IOIS, including orbital myositis, are rare. Avni-Zauberman et al. [21] reported a case series with recurrent migratory IOIS involving different EOM. However, they reported only which muscles were involved without a detailed description on ocular motility.

The purpose of our study was to present the detailed clinical features and patterns of ocular motility disturbance in a relatively large number of patients with IOIS, which, as far as we know, have not been published in the literature. We also examined risk factors for recurrence of orbital myositis and compared them with previously published studies.

## 2. Materials and Methods

A retrospective review was performed on 31 patients who visited the Department of Ophthalmology, Seoul National University Bundang Hospital between the years 2003 to 2019 and were confirmed to have EOM affected by idiopathic orbital inflammation. Patients who were diagnosed as having thyroid orbitopathy and IgG4-related ophthalmic diseases (ROD), based on the diagnostic criteria for definite or probable IgG4-ROD proposed by Goto et al. [22], were excluded. Patients with cranial nerve III, IV, or VI palsy, Brown syndrome, Duane retraction syndrome, other restrictive strabismus, or unilateral EOM fibroma localized to one muscle were also excluded. The study protocol was approved by the institutional review board of Seoul National University Bundang Hospital (B-1612/374-102).

Ophthalmologic examinations including visual acuity, ocular alignment with alternate prism cover tests at 6 cardinal gazes, ductions and versions, pupillary examination, slit lamp examination, and fundus examinations were performed. EOM limitation was quantitatively scored for upgaze, downgaze, abduction, and adduction by the severity of limitation on a grading scale of 0 to −4. A grade of 0 was noted for full excursion, −4 for 0% excursion just reaching midline, and −3 to −1 for 25% increments. In cases with multiple EOM involvement, we regarded the muscle that showed the most severe limitation in duction as the most affected muscle. If the grade of limitation was the same, we considered the one having more enlargement in size and higher contrast enhancement compared to the opposite eye. Finally, the sum of the limitation grades in each direction was calculated. The degree of EOM limitation was analyzed only when muscle enhancement or enlargement was clearly observed on CT or MR imaging.

CT imaging was conducted using detector-row machines (Philips Medical Systems, Cleveland, OH, USA) with an intravenous nonionic contrast material (2 mL/kg; iopromide, Ultravist 370; Bayer, Berlin, Germany). Axial and coronal images were reconstructed with 2 mm thickness at 3 mm intervals. MR imaging was conducted using a 1.5 tesla system (Gyroscan Intera; Philips, Healthcare, Best, The Netherlands) or a 3 tesla system (Achieva; Philips, Healthcare, Best, The Netherlands) with a SENSE (SENSitivity Encoding) head coil. T1- and T2-weighted imagings were performed to evaluate the orbit including EOM. Abnormalities in orbital contents including the EOM were reviewed.

All patients were initially treated with corticosteroids and checked for symptom improvement after at least 5 weeks of treatment, as response to treatment is usually evident by this time. The initial dose of oral prednisolone was determined by the patient’s weight as 0.5–1 mg/kg/day. The tapering schedule depended on the patient’s clinical symptoms. If symptoms improved, the daily dose was decreased by 5–10 mg per every week. Each patient was evaluated at least three times. Treatment outcome was considered a “complete resolution and cure” if the patient had complete relief of symptoms for at least 3 months or more without recurrence after stopping medication. “The recurrence of each episode of orbital myositis” was defined as one of the following findings: (1) relapse of eye symptoms including pain with eye movement or limitation of ocular motility, and/or (2) change in EOM size and enhancement on orbital imaging. According to the definition mentioned above, we classified patients into a ‘cured group’ and a ‘recurred group’. In addition, the recurred group was divided into patients with multiple recurrences of a total of more than 3 attacks, and those with a single recurrence and no extra recurrence after the second attack.

## 3. Results

### 3.1. Clinical Characteristics

The mean age was 43.8 ± 14.4 (range, 12–78 years). Among them, 17 patients (55%) were males and 14 (45%) were females. The mean follow-up duration was 79.8 ± 96.8 weeks (range, 5–403 weeks).

Twenty-six patients (84%) had unilateral involvement and five patients (16%) were bilateral. Diplopia was the most frequent symptom in 14 patients (45%) and 12 patients (39%) complained of ocular pain. Only 5 patients (16%) appealed visual disturbance. The most common sign was eyelid swelling in 13 patients (42%), followed by proptosis (20%), ptosis (3%), and conjunctival injection (3%).

The characteristics of muscle involvement are summarized in Table 1. Single muscle involvement was found in 13 patients (42%), while the other 18 patients (58%) had two or more muscles involved. Lacrimal gland involvement was found in three patients (10%) accompanied by orbital myositis resulting in limitation of eye movements. The most frequently involved muscle was the medial rectus (37%), followed by the lateral rectus (23%), inferior rectus (20%), superior rectus (18%), and superior oblique (2%).

The mean grade of total EOM limitation was −1.65 ± 1.80 (range, −7 to 0). There was no limitation of ocular movement in 9 patients (29%), while 22 patients (71%) showed variable degrees of ocular motility limitation. EOM limitation was found in the same direction of the most affected muscle in 14 patients (64%), while 8 patients (36%) complained of duction limitation in the opposite direction. Among the 14 patients who showed EOM limitation towards the field of action of the affected muscle, the most frequently involved muscle was the lateral rectus (n = 6) and the superior rectus (n = 6), followed by medial rectus (n = 2) and inferior rectus (n = 2). The grade of EOM limitation was most severe in the medial rectus (−2.50 ± 0.71), followed by the inferior rectus (−2.00 ± 1.41), lateral rectus (−1.75 ± 1.17), and superior rectus (−1.58 ± 0.80) muscles. On the other hand, among the 8 patients who showed EOM limitation in the opposite direction of the affected muscle, the most frequently involved muscle was the medial rectus (n = 4) followed by the superior rectus (n = 3) and inferior rectus muscle (n = 1). The grade of EOM limitation was most severe in the superior rectus (−1.67 ± 0.58), followed by the medial rectus (−1.38 ± 0.48) and lateral rectus muscle (−1.00 ± 0.00) (Table 2).

### 3.2. Treatment Outcome

All patients took systemic steroids, 23 patients (74%) were treated with oral prednisolone alone and the average initial dose was 48.2 ± 18.5 mg/day (range, 20 to 80 mg/day). The mean interval from the application of the first dose of steroid until the first noticeable relief of initial symptoms was 3.3 ± 2.3 weeks (range, 1–12 weeks). Five patients (16%) required treatment with additional immunosuppressant drugs including methotrexate and azathioprine. One patient (3%) was treated with oral steroids, immunosuppressants, and intravenous immunoglobulin. In two patients (7%), symptoms resolved without any treatment.

Excluding five patients whose follow-up period was shorter than three months after stopping medication, 26 patients were evaluated for recurrence. Nine patients (35%) suffered from recurrence (recurred group) while 17 patients (65%) had complete resolution of signs and symptoms without recurrence (cured group). The initial dose of steroids between the cured group (47.7 ± 22.4 mg/day, 0–80) and recurred group (47.2 ± 14.4 mg/day, 20–70) was not significantly different (*p* = 0.490). The mean duration of initial steroid treatment was 14.6 ± 17.3 weeks (range, 0–77 weeks) in the cured group and 26.6 ± 41.6 weeks (range, 8–116 weeks) in the recurred group (*p* = 0.588). Recurred patients were more likely to have multiple muscle involvement; the mean number of involved muscles was 1.76 ± 0.83 (range, 1–3) in the cured group and 3.22 ± 0.67 (range, 2–5) in the recurred group (*p* < 0.001).

### 3.3. Relapsing Cases

The characteristics of the nine patients with recurrence are summarized in Table 3. Six patients were males and three patients were females. Among these nine patients, eight patients presented with recurrent symptoms that were similar to the initial symptoms. The remaining one patient presented with a new onset of exophthalmos and pain during recurrence (Case 9). Follow-up orbital imaging performed during recurrence showed that six patients recurred in the same EOM that were initially involved, while three patients showed new inflammatory lesions that were not initially involved. Among the seven patients with EOM limitation, four patients showed EOM limitation to the opposite direction of the involved muscle both at initial onset and during recurrence, suggesting restriction (Case 2, 4, 7, and 8). On the other hand, three out of the 7 patients with EOM limitation showed different patterns of muscle involvement at onset and during recurrence, and EOM limitation was prominent towards the action of the involved muscle, suggesting muscle palsy (Case 1, 6, and 9).

All relapsed patients took oral steroids, and two of them received additional methotrexate. The average duration of using steroids for retreatment was 13.1 ± 8.7 weeks (range, 1–29 weeks). The mean interval to relapse of symptoms after discontinuation of steroids was 4.6 ± 2.4 weeks (range, 1–8 weeks). Of the 9 recurred patients, 6 patients fully recovered after retreatment, but one patient had a third attack and the remaining two patients showed chronic progressive courses with relapse and remission more than four times. In particular, the interval to relapse of symptoms after discontinuation of steroids was significantly shorter in patients with multiple recurrences compared to those with a single recurrence (1.8 ± 0.8 weeks versus 6.0 ± 1.4 weeks, respectively, Mann–Whitney U-test, *p* = 0.020).

## 4. Discussion

In this study, we analyzed the clinical characteristics and ocular motility of 31 patients with idiopathic orbital inflammatory syndrome. Of 22 patients with EOM limitation, 14 patients (64%) developed EOM limitation in the same direction of the most affected muscle. Of the 26 patients who were followed up for more than 3 months, 17 patients (65%) showed complete resolution without recurrence while nine patients (35%) recurred during follow-up. We found that multiple muscle involvement was likely to contribute to recurrence in orbital myositis. In addition, a rapid relapse of symptoms after discontinuation of steroids was a significant indicator of multiple recurrences.

Orbital myositis is regarded as a subtype of IOIS [23,24,25]. IOIS usually manifests in the fifth decade and there is no sex predilection [1]. However, orbital myositis most commonly affects young adults in the third to fourth decade and shows more prevalence in women [26,27]. In our patients, the mean age was 43.8 in the fifth decade, and the male to female ration was 17:14, similar to previous reports. Orbital myositis is rare in the pediatric age group, accounting for only 6% to 17% of the total incidence, which is similar to our study (7%) [25,28,29]. There were only two children in our study, who were 12 and 14 years old, respectively. One patient had bilateral disease and recurrent myositis (Case 3). However, no significant difference from adults was found due to the small number of patients.

The pattern of EOM involvement in our study was similar to the previous literature [1,30]. It was reported that orbital myositis was a unilateral process involving a single EOM in the initial stage [31,32]. However, recent studies have shown that myositis often affects more than one muscle and is frequently bilateral [30]. In our study, multiple EOM were affected in 18 patients (58%). Yuen et al. [1] described EOM involvement in 26 patients with orbital myositis. They reported that the most frequently involved muscle was the medial rectus (n = 18; 31%), followed by the superior rectus (n = 15; 25%), lateral rectus (n = 14; 24%), and inferior rectus (n = 12; 20%). Oblique muscle paralysis has been reported together with multiple muscle involvement [33]. However, we found oblique muscle involvement in only one (1.7%) patient (Figure 1).

In our study, the relationship between gaze limitation/pain and the involved EOM was not always consistent. For example, in our study, myositis of the superior rectus muscle presented with pain and diplopia on upgaze in Case 6 and 9, while Case 8 had pain and diplopia during downgaze. Generally, in orbital myositis, periocular pain and diplopia are aggravated by eye movement towards the involved muscle [5,30,33]. This is probably caused by limited contraction of the affected EOM [30]. We might regard this type of muscle involvement as the “palsy type”. On the other hand, as seen in 8 patients in our study, EOM limitation may be prominent towards the opposite or different direction from the field of action of the affected muscle observed on CT or MR imaging. This finding implies the complexity of eye muscle coordination where the time of onset and degree of fibrosis may have partially affected eye movement [34]. EOM limitation can result from restrictive disease of the orbit or from abnormal muscles, and several studies have reported limitation of eye movement due to EOM fibrosis in patients with IOIS [35]. Likewise, we could consider this type of EOM limitation as the “restriction type”. The general understanding of an acute inflammation does not commonly imply an immediate yield of fibrosis which is considered as a chronic response. In the cases with a suspected “restrictive” pattern, we could assume that a subacute process of chronic inflammation has been ongoing, and inflammatory involvement of the respective antagonist muscle could be excluded with CT or MR imaging. In our study, all four patients with recurrence in the same muscle showed EOM limitation to the opposite direction of the involved muscle both at onset and during relapse. This suggests a “restriction type” due to muscle fibrosis rather than a “palsy type”. Based on these patients, we might assume that the reason for poor response to treatment is partly due to severe muscle fibrosis. The forced duction test (FDT) may confirm the presence of mechanical restriction. However, we did not perform the FDT in most of the patients in our study due to their presentation of EOM pain and acute inflammation, which is one of the limitations in revealing the exact mechanism of EOM limitation in IOIS. Regarding orbital MR imaging protocols in our institution, we perform fat suppression using fluid attenuated inversion recovery (FLAIR), Spectral Presaturation with Inversion Recovery (SPIR), and Dixon methods to evaluate inflammation in a manner similar to the short tau inversion recovery (STIR) sequence. Overall, a slightly higher signal intensity was seen in the palsy type compared to the restriction type. However, not all patients were tested with MRI or the same sequence protocol, and we do not know for sure whether there is a remarkable difference that can be clinically estimated by MR imaging alone. Further studies are required to investigate this issue.

Several studies have reported on the outcome of orbital myositis. Mannor et al. [5] described recurrence of orbital myositis in 6 out of 26 patients (23%) and reported that several characteristics are associated with recurrent orbital myositis: male gender, eyelid retraction, lack of proptosis, horizontal recti inflammation, multiple muscle inflammation, bilateral muscle inflammation, tendon sparing, and lack of response to nonsteroidal anti-inflammatory drugs or corticosteroids. Lacey et al. [27] pointed out that multiple muscle involvement at the initial presentation was a risk factor for recurrence, particularly if bilateral. Similarly, in our study, multiple muscle inflammation was associated with relapse of orbital myositis. Recurrence of orbital myositis is common in the same eye during the course of the disease, but rare in the opposite eye [6,36]. However, in our study, three out of 9 patients (33%) showed new inflammatory lesions during recurrence that were not initially involved, and two patients (22%) particularly recurred in the opposite eye. On the other hand, there might be a possibility that rapid tapering of steroids could have affected early relapse and multiple recurrences in idiopathic orbital myositis. However, in our study, the mean duration of initial steroid treatment was 6.6 ± 10.2 months in the recurred group, which is longer than other studies (2.3 ± 1.5 months) [36] or the recommended treatment period of 6 to 12 weeks [26,30,37,38].

This study has a few limitations. First, it was a retrospective case series including a small number of patients and variable follow-up durations. Second, biopsy was performed in only one patient with chronic progression while three patients with multiple recurrences did not undergo biopsy to rule out other causes as the response to steroid was excellent and disease progression was not definite.

In conclusion, we analyzed the detailed features of ocular motility and risk factors for recurrence in IOIS. Patients showed variable degree of ocular motility limitation, and EOM limitation was more frequently found in the same direction of the affected muscle as a palsy type. Patients with recurrent myositis were more likely to have multiple muscle involvement. A rapid relapse of symptoms after discontinuation of steroids was a significant indicator of multiple recurrences.

## Figures and Tables

**Figure 1 jcm-09-01165-f001:**
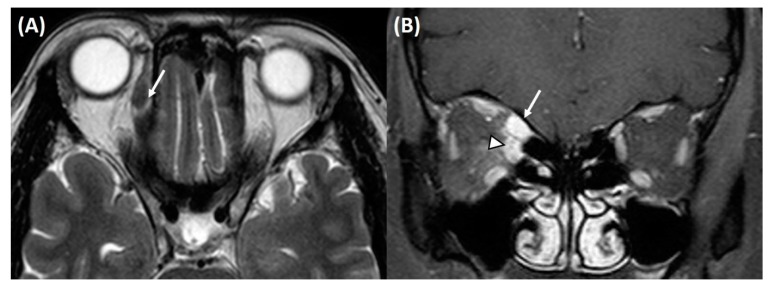
(**A**) T2-weighted axial and (**B**) T1-weighted coronal magnetic resonance imaging with gadolinium contrast shows enlargement and contrast enhancement of the right superior oblique (white arrow) and medial rectus muscles (white arrowhead).

**Table 1 jcm-09-01165-t001:** The characteristics of muscle involvement.

	Muscle	Number (%)
Involved patterns * (Patient No.)	4 muscles or more	5 (16.7)
	3 muscles	5 (16.7)
	2 muscles	8 (25.8)
	1 muscle	13 (41.9)
	Lacrimal gland involvement	3 (9.7)
Involved frequency * (Muscle No.)	Medial rectus	22 (36.7)
	Lateral rectus	14 (23.3)
	Inferior rectus	12 (20.0)
	Superior rectus	11 (18.3)
	Superior oblique	1 (1.7)

* including multiple involvement.

**Table 2 jcm-09-01165-t002:** Analysis of extraocular muscle (EOM) limitation by involved muscles.

Direction of EOM Limitation	Muscle	Number	Grading Scale of EOM Limitation (Mean ± SD)
In the same direction of the most affected muscle (palsy type, 14 patients)	Lateral rectus	6	−1.75 ± 1.17
	Superior rectus	6	−1.58 ± 0.80
	Medial rectus	2	−2.50 ± 0.71
	Inferior rectus	2	−2.00 ± 1.41
In the opposite direction of the most affected muscle (restriction type, 8 patients)	Medial rectus	4	−1.38 ± 0.48
	Superior rectus	3	−1.67 ± 0.58
	Inferior rectus	1	−1.00 ± 0.00

**Table 3 jcm-09-01165-t003:** Clinical characteristics of 9 patients with relapsing orbital myositis.

Case	Initial Symptoms	Abnormal Imaging (1st Episode)	EOM Limitation (1st Episode)	Abnormal Imaging (2nd Episode)	EOM limitation (2nd Episode)	Interval to Relapse * (Weeks)	Total No. of Attacks	Total Period of Treatment (Weeks)	Acute Phase Treatment
1	Lid swelling, Pain	LMR	L) Add −1.5	LSR	L) Up −1	5	2	37	PD 50 mg
2	Diplopia, Pain	RMR, RSO, RSR	R) Down −2	RMR, RSR	R) Down −3, Add −4	2.5	> 4	39	PD 55 mg MTX 2.5 mg
3	Lid swelling	LMR, LLR, LIR, Lt. lacrimal, RMR	-	LMR, LLR, LIR, Lt. lacrimal	-	7	2	13	PD 40 mg
4	Diplopia, VA loss	RSR, RMR, RIR, LSR	R) Down −1	RSR, LIR	R) Down −2, L) Down −4	2	3	31	PD 70 mg
5	Proptosis	LMR, LIR, LLR, Lt. lacrimal	-	LLR, Lt. lacrimal	-	4	2	8	PD 40 mg
6	Diplopia, Lid swelling	LSR, LLR	L) Abd −4, Up −1	RLR	R) Abd −1	6	2	19	PD 50 mg
7	Diplopia, Pain	LSR, RMR	R) Abd −2	RMR	R) Abd −2	1	> 4	148	PD 40 mg MTX 15 mg
8	Diplopia, Pain	LSR, LLR	L) Down −2	LSR, LLR	L) Down −2	6	2	35	PD 20 mg
9	Diplopia	LSR	L) Up −3	RIR	R) Down −1	8	2	27	PD 60 mg

* interval to relapse of symptoms after discontinuation of steroids. EOM: Extraocular Muscles, LMR: Left Medial Rectus, LLR: Left Lateral Rectus, LSR: Left Superior Rectus, LIL: Left Inferior Rectus RMR: Right Medial Rectus, RLR: Right Lateral Rectus, RSR: Right Superior Rectus, RIR: Right Inferior Rectus, RSO: Right Superior Oblique, Lt: left, MTX: Methotrexate, Abd: Abduction, Add: Adduction, PD: oral prednisolone, VA: visual acuity

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
