# Peer review of "Clinical Features of Ocular Motility in Idiopathic Orbital Myositis"

_jcm, 2020, doi:10.3390/jcm9041165_

Round 1

Reviewer 1 Report

This paper describes multiple recurrence of orbital myositis can be predicted from rapid relapse of symptoms. It is a bit disappointing that their way of treatment and the length of follow-up is not uniform with its retrospective design. The recurrence may increase as the follow-up is extended.

The authors should address the following.

  1. Materials and Methods. The definition of recurrence should be clarified.
  2. Materials and Methods. Please add the description about how initial dose of steroids and tapering schedule were determined.
  3. Table 1. Medical is misspelled.
  4. Results (Clinical Characteristics) How to evaluate “most affected muscles” in cases with two or more muscle involvement should be clarified.
  5. Results (Relapsing cases) patents is misspelled.
  6. Results (Relapsing cases) Please add the number of the case after “the remaining one patient”.
  7. MR using the Short Tau Inversion Recovery (STIR) sequence is helpful in evaluating the degree of inflammation of extraocular muscles. Are there any difference of STIR image between “palsy type” and “restriction type”?
  8. Please discuss about the relationship of complain of ocular pain and CT or MR imaging.
  9. In Vogt-Koyanagi-Harada disease, the recurrence of inflammation is not uncommon after cessation of corticosteroid therapy, especially when the steroid is tapered too early. (Lai TY et al: Eye, 2009, 23; 543-548) Similarly, is there a possibility that too early tapering of the steroid affects a rapid relapse and multiple recurrences in idiopathic orbital myositis? Please discuss this point.

Author Response

I thank you for taking your time to review our manuscript. A response to your comments/requests in accordance with the recommendation is included below and upload it as a Word file. I hope the revised manuscript will better meet the publication requirement of JCM.

  1. Materials and Methods. The definition of recurrence should be clarified.

⇒ Yes, we added additional explanation in the materials and methods section as follows: page 2, section Material and Methods, last paragraph

“The recurrence of each episode of orbital myositis was defined as one of the following findings; 1) relapse of eye symptoms including pain with eye movement or limitation of ocular motility, and/or 2) change in EOM size and enhancement on orbital imaging.”

  1. Materials and Methods. Please add the description about how initial dose of steroids and tapering schedule were determined.

⇒ Yes, we added additional explanation in the materials and methods section as follows: page 2, section Material and Methods, last paragraph

“The initial dose of oral prednisolone was determined by the patient's weight as 1mg/kg/day. The tapering schedule depended on the patient's clinical symptoms. If symptoms improved, the daily dose was decreased by 10 mg per every week.”

  1. Table 1. Medical is misspelled.

⇒ Thanks, we corrected it.

  1. Results (Clinical Characteristics) How to evaluate “most affected muscles” in cases with two or more muscle involvement should be clarified.

⇒ Yes, we added additional explanation in the materials and methods section as follows: page 2, section Material and Methods, the second paragraph

“We regarded the muscle that showed the most severe limitation in duction as the most affected muscle. If the grade of limitation was the same, we considered the one having more enlargement in size and higher contrast enhancement compared to the opposite eye.”

  1. Results (Relapsing cases) patents is misspelled

⇒ Thanks, we corrected it.

  1. Results (Relapsing cases) Please add the number of the case after “the remaining one patient”.

⇒ Yes, we added the number, case 9.

  1. MR using the Short Tau Inversion Recovery (STIR) sequence is helpful in evaluating the degree of inflammation of extraocular muscles. Are there any difference of STIR image between “palsy type” and “restriction type”?

⇒ Yes, we added additional explanation in the discussion section as follows: page 6-7, section Discussion, last paragraph

“In our institution, we perform fat suppression using FLAIR, SPIR (Spectral Presaturation with Inversion Recovery) and DIXON methods to evaluate inflammation in a manner similar to the STIR sequence. Overall, a slightly higher signal intensity was seen in the palsy type compared to the restriction type. However, not all patients were tested with MRI or the same sequence protocol, and we do not know for sure whether there is a remarkable difference that can be clinically estimated by MR imaging alone. Further studies are required to investigate this issue.”

We added MR imaging of some patients included in our study as follows.

-> Please check the attached Word file.

  1. Please discuss about the relationship of complain of ocular pain and CT or MR imaging.

⇒ Yes, we added additional explanation in the discussion section as follows: page 6, section Discussion, last paragraph

“In our study, the relationship between gaze limitation/pain and the involved EOM was not always consistent. For example, in our study, myositis of the superior rectus muscle presented with pain and diplopia on upgaze in case 6 and 9, while case 8 had pain and diplopia during downgaze.”

  1. In Vogt-Koyanagi-Harada disease, the recurrence of inflammation is not uncommon after cessation of corticosteroid therapy, especially when the steroid is tapered too early. (Lai TY et al: Eye, 2009, 23; 543-548) Similarly, is there a possibility that too early tapering of the steroid affects a rapid relapse and multiple recurrences in idiopathic orbital myositis? Please discuss this point.

⇒ Yes, we added additional explanation in the discussion section and added more references as follows: page 7, section Discussion, the second paragraph

“On the other hand, there might be a possibility that rapid tapering of steroids could have affected early relapse and multiple recurrences in idiopathic orbital myositis. However, in our study, the mean duration of initial steroid treatment was 6.6 ± 10.2 months in the recurred group, which is longer than other studies (2.3 ± 1.5 months) [36] or the recommended treatment period of 6 to 12 weeks [26,30,37,38].”

Reviewer 2 Report

The manuscript is well written, properly organized, with a clear and brief Introduction in which the purpose of the study is well detailed. Results are also well presented, both in text and in tables. Discussion is brief and clear.

I would just like to do 2 comments:

  • Typo in the word “recurrence” (page 2, section Material and Methods, last paragraph)
  • It would be useful if the authors made a particular comment regarding the pediatric age subset. Based on previous reports, there is an impression that presentation in children is more frequently bilateral than in adults. It would be useful to know if this was also true in this case series, and if other clinical characteristics and response to treatment differed from the adult subgroup.

Author Response

I thank you for taking your time to review our manuscript. A response to the your comments/requests in accordance with the recommendation is included below and upload it as a Word file. I hope the revised manuscript will better meet the publication requirement of JCM.

1. Typo in the word “recurrence” (page 2, section Material and Methods, last paragraph)

⇒ Thanks, we corrected it.

2. It would be useful if the authors made a particular comment regarding the pediatric age subset. Based on previous reports, there is an impression that presentation in children is more frequently bilateral than in adults. It would be useful to know if this was also true in this case series, and if other clinical characteristics and response to treatment differed from the adult subgroup

⇒ Yes, we added additional explanation in the discussion section as follows: page 5, section Discussion, last paragraph

“There were only two children in our study, who were 12 and 14 years old, respectively. One patient had bilateral disease and recurrent myositis. (Case 3) However, no significant difference from adults was found due to the small number of patients.”

Thank you very much for your comments.
